# Perception in the Dark; Development of a ToF Visual Inertial Odometry System

**DOI:** 10.3390/s20051263

**Published:** 2020-02-26

**Authors:** Shengyang Chen, Ching-Wei Chang, Chih-Yung Wen

**Affiliations:** Deptartment of Mechanical Engineering and Interdisciplinary Division of Aeronautical and Aviation Engineering, The Hong Kong Polytechnic University, Kowloon, Hong Kong; shengyang.chen@connect.polyu.hk (S.C.); chingwei.chang@connect.polyu.hk (C.-W.C.)

**Keywords:** VIO, ToF camera, real time, error-state Kalman Filter, data fusion, ICP

## Abstract

Visual inertial odometry (VIO) is the front-end of visual simultaneous localization and mapping (vSLAM) methods and has been actively studied in recent years. In this context, a time-of-flight (ToF) camera, with its high accuracy of depth measurement and strong resilience to ambient light of variable intensity, draws our interest. Thus, in this paper, we present a realtime visual inertial system based on a low cost ToF camera. The iterative closest point (ICP) methodology is adopted, incorporating salient point-selection criteria and a robustness-weighting function. In addition, an error-state Kalman filter is used and fused with inertial measurement unit (IMU) data. To test its capability, the ToF–VIO system is mounted on an unmanned aerial vehicle (UAV) platform and operated in a variable light environment. The estimated flight trajectory is compared with the ground truth data captured by a motion capture system. Real flight experiments are also conducted in a dark indoor environment, demonstrating good agreement with estimated performance. The current system is thus shown to be accurate and efficient for use in UAV applications in dark and Global Navigation Satellite System (GNSS)-denied environments.

## 1. Introduction

The study is motivated and inspired by the need to design and build an efficient and low-cost robot-mount localization system for indoor industrial facility inspection, tunnel inspection, and search and rescue missions inside earthquake- or explosion-damaged buildings in poor lighting and GNSS-denied conditions. The system must be able to track the movement of the inspection vehicle in real time without the assistance of GNSS, GPS or other localization infrastructure. A vision based state estimator is particularly suitable for such a scenario.

Consequently, vision-based state estimation has become one of the most active research areas in the past few years and there have been many notable VO, VIO or VSLAM works, such as parallel tracking and mapping (PTAM) [1], SVO+MSF (semidirect visual odometry + multiple sensor fusion) [2,3,4], MSCKF (multi-state constrain Kalman filter) [5], OKVIS (open keyframe-based visual–inertial SLAM) [6], VINS (visual–inertial navigation system) [7,8], VI-ORB-SLAM (visual–inertial oriented FAST and rotated BRIEF-SLAM) [9], VI-DSO (visual–inertial direct sparse odometry) [10] and KinectFusion [11]. These works can be categorized by their pose estimation methods and the ways of fusing IMU data (Table 1). Currently, approaches to depth information based pose estimation can generally use one of two optimization methods: the direct optimization method and the iterative closest point (ICP) method.

The optimization method models the estimation task as a minimization problem [15,16]. Generally, two kinds of optimization methods are widely used: Image-based optimization method and direct optimization method. In the traditional image-based optimization method, the objective function is modeled by the re-projection error of the features through feature matching; while in the direct optimization method, with the photometric camera model, the objective function is chosen as the intensity residual between frames. A comprehensive comparison of these two optimization methods can be found in Delmerico and Scaramuzza [17]. The target functions are solved by the gradient descent optimizer. As the image transformation in the traditional image-based method is represented by the Jacobian matrix in the objective function, which is the first order linearization of the real model, this method performs poorly in scenarios involving significant rotation, due to the implicitly high non-linearity in the real model.

Apart from the traditional image-based optimization method, the ICP method is considered as a relatively new type of its kind. The ICP method relies on the classic registration workflow [18], where the ICP algorithm begins with two sequential point clouds and an initial prediction of the transformation. The correspondence between two images (i.e., the source point cloud and the target point cloud) is found by searching the closest neighborhoods of the source points and estimating the progressive transformation to minimize the distances between the source points and target points. The iteration process is continued until transformation convergence is achieved. The value of the initial guess for the ICP is important, as a poor guess may lead to the iterative solution becoming trapped in a local minimum. Also, the ICP require high computation power especially when the number of points increase.

Additionally, there are some others, such as normal distribution transformation (NDT), in which the transform is calculated based on the probability density function inside the celled map. This kind of methods require huge amount of depth data and is normally used in the Lidar sensor applications. The notable works using the NDT method include: 3D-NDT [13,19] and Direct Depth SLAM [14].

Most of the methods described above use passive cameras and were developed for use under different experimental environments in daylight, meaning that camera exposure for good image quality is not a problem. However, many real application scenarios, such as indoor industrial facility inspection and search and rescue missions inside earthquake- or explosion-damaged buildings, are much more challenging than the benchmark case of the MH 05 difficult in EuRoC MAV Dataset [20], as these are in poor lighting conditions. In order to achieve perception in such dark or changing ambient light environments, we replace the conventional passive camera with an active ToF camera in this study. The pros and cons of using the conventional passive camera and the active ToF camera in motion estimation are listed in Table 2.

Notably, in a passive camera, the sensor is exposed to the light during the shutter opening period. The depth information can be triangulated through the motion of the camera or using two cameras with the calibrated extrinsic parameters. Contrarily, the ToF camera illuminates a modulated light and observes the reflected light. The phase shift between the illuminating light and the reflected light is measured and translated to distance [21]. Since such a ToF camera can measure the distance in “one shot” without any post processing, it enables good resilience to any change of environmental brightness [22]. The output of the ToF camera provides a depth image and an NIR intensity image simultaneously (Figure 1).

In this paper, we propose an optimized VIO framework using a combination of a pmd flexx ToF camera and Pixhawk v2 hardware as the IMU sensor. Salient point selection with a statistic based weight factor is applied in the basic ICP algorithm, and the estimation results are fused with the IMU data using an error state Kalman filter. The algorithm can achieve a realtime processing rate for UAV applications without needing to use a graphics-processing unit. The main contributions of this work are:Improved the conventional ICP workflow with the salient point selection criterias and the statistics based robust weighting factor.Development of an error state Kalman filter based framework to fuse global sensors, composed of a ToF camera and an IMU, and with local estimations, which achieves locally accurate localization and high computational efficiency.An evaluation of the proposed system in both the motion capture system and the real experiments. A ToF-IMU dataset with the ground truth is published.Open-source code of the error state Kalman filter based ToF–VIO for the research community.

## 2. ToF–VIO Platform

Our sensor platform consists of a pmd technologies Flexx ToF camera and an IMU sensor embedded in a Pixhawk v2 flight controller, as shown in Figure 2. The Flexx ToF camera has a resolution of 224 × 171 pixels and a depth detection range of 4 m with a frame rate set to be 15 Hz. The Pixhawk v2 IMU updates at a rate of 250 Hz. The times of the two sensors are synchronized by the software and the installation geometry of the camera and IMU is represented by (Equation 1).
(1)$$T_{IC}=\left[\begin{array}{cccc} 0 & 0 & 1 & 0.1\\ -1 & 0 & 0 & 0\\ 0 & -1 & 0 & 0\\ 0 & 0 & 0 & 1 \end{array}\right]$$

## 3. ICP Alignment

### 3.1. Visual Odometry by the ToF Camera Module

The ToF camera outputs a depth image $p_{z}=\left(u,v,z\right)$ and a near infrared (NIR) intensity image $p_{i}=\left(u,v,i\right)$ simultaneously, where *z* and *i* represent the depth and intensity information, respectively, of the pixel (u,v). Because these two images are captured by the same optical sensing module and aligned, they share the same camera parameters ($f_{x},c_{x},f_{y},c_{y}$), where *f* and *c* are the focal length and the image center, respectively. For simplicity, we can use p=(u,v,z,i) to represent the camera output, which is linked to its corresponding 3D point in the Camera Frame with the intensity, $P_{c}={\left(X_{c},Y_{c},Z_{c},I\right)}^{T}$, through the projection model of the camera as shown in Equation (Equation 2). Accordingly, $P_{c}=\pi \left(p\right)$ can be derived explicitly in Equation (Equation 3).
(2)$$\left[\begin{array}{c} u\\ v\\ 1\\ z\\ i \end{array}\right]=\left[\begin{array}{ccccc} f_{x} & 0 & c_{x} & 0 & 0\\ 0 & f_{y} & c_{y} & 0 & 0\\ 0 & 0 & 1 & 0 & 0\\ 0 & 0 & 0 & 1 & 0\\ 0 & 0 & 0 & 0 & 1 \end{array}\right]\left[\begin{array}{c} X_{c}/Z_{c}\\ Y_{c}/Z_{c}\\ 1\\ Z_{c}\\ I \end{array}\right]$$
(3)$$P_{c}=\pi \left(p\right)={\left(\frac{u\cdot f_{x}}{z}+c_{x},\frac{v\cdot f_{y}}{z}+c_{y},z,i\right)}^{T}$$

Assuming the world frame is fixed, the rigid motion of an object in front of a still camera can be described by a transformation matrix T=[R|t], where R is the rotational matrix in SO(3) and *t* represents the translational vector. Equation (Equation 4) describes such a transformation from the world frame to the camera frame, where $P_{w}={\left(X_{w},Y_{w},Z_{w},I\right)}^{T}$ is the 3D point in the world Frame with its intensity.
(4)$$P_{c}=\left[\begin{array}{c} X_{c}\\ Y_{c}\\ Z_{c}\\ I \end{array}\right]=T\cdot P_{w}=\left[\begin{array}{cccc} r_{11} & r_{12} & r_{13} & t_{x}\\ r_{21} & r_{22} & r_{23} & t_{y}\\ r_{31} & r_{32} & r_{33} & t_{z}\\ 0 & 0 & 0 & 1 \end{array}\right]\left[\begin{array}{c} X_{w}\\ Y_{w}\\ Z_{w}\\ I \end{array}\right]$$

As shown in Figure 3, a moving camera captures the point clouds $P_{t1}$ and $P_{t2}$ at two sequent time t1 and t2, respectively. Notably, P and *P* represent the point cloud and a single point, respectively. The captured point cloud, $P_{t1}$ can be represented by Equation (Equation 5), while $P_{t2}$ can be regard as the inter-frame transformation of $P_{t1}$ due to the motion of the camera in the world frame, as seen in Equation (Equation 6). Here, $T_{cam}$ represents the coordinate transformation of the camera in the world frame between $P_{t1}$ and $P_{t2}$.
(5)$$P_{t1}=T_{t1}\cdot P_{w}$$
(6)$$P_{t2}=T_{cam}^{-1}P_{t1}=T_{cam}^{-1}\cdot T_{t1}\cdot P_{w}$$

The frame to frame alignment can also be made through the transformation matrix T, which aligns two point clouds as $P_{t2}=T\cdot P_{t1}$. Combining it with Equation (Equation 5) and Equation (Equation 6), we could find the motion of the camera $T_{cam}$ equal to the inverse of the point cloud transformation matrix T (Equation (Equation 7)). In another word, the camera motion can be derived from the point cloud alignment.
(7)$$P_{t2}=T_{cam}^{-1}\cdot T_{t1}\cdot P_{w}=T\cdot T_{t1}\cdot P_{w}\Rightarrow T_{cam}=T^{-1}$$

In this work, the ICP algorithm are used to derive the transformation T. The workflow of the basic ICP algorithm can be summarized as:The point cloud alignment algorithm starts with the source point cloud $P_{s}$ and the target point cloud $P_{t}$ and an initial prediction of the transformation $T_{0}$ between these two clouds.For every point in the target cloud ($P_{i}\in P_{t}$), searching the corresponding point $P_{i}^{\prime }$ in the transformed source point cloud which has the closest distance:
(8)$$P_{i}^{\prime }=P_{j}\;,\;j=\underset{j}{\mathrm{argmin}}\left|\right|P_{i}-T\cdot Pj\left|\right|\;,\;P_{j}\in P_{s}$$Estimate the increment transformation from the point pairs which could minimize the error metric.
(9)$$\Delta T=\underset{\Delta T}{\mathrm{argmin}}||P-\Delta T\cdot P^{\prime }\left|\right|$$Applying the increment transformation.
(10)$$T^{n+1}\leftarrow \Delta T\cdot T^{n}$$By applying the above steps iteratively until the transformation of the two point clouds converge (Equation (Equation 11), where $T_{TH}$ is the threshold of transformation) or meets a certain criterion, e.g reach maximum iterative number, the final transformation is obtained.
(11)$$\Delta T<T_{TH}$$

Many variants of the basic ICP concept have been introduced to enhance the performance, i.e, to speed up the algorithm, improve robustness or increase accuracy. In the work of Rusinkiewicz and Levov [23], they classify these variants according to the effect on one of six stages of the algorithm: Selection, Matching, Weighting, Rejecting, Assigning Error Metric and Minimizing the error metric. Inspired by the work of Li and Lee [24], we develop a salient point selection criteria for acceleration of the ICP process. The statics based weighting function to improve the robustness of the ICP process is applied.

### 3.2. Selection of Sailent Points

The resolution of ToF camera in this study is 38K, i.e, every point cloud image contains 38K points. It is impossible to align such a huge amount of points on an embedded computer in real time. Therefore, several criteria to select the salient points from the raw image are applied in this work. As shown in Figure 4, these criteria can be divided into the rejection and acceptance groups.

#### 3.2.1. Rejection Rules

Rejection Rules: Two kinds of rejection rules are applied: background points and motion induced outliers. A cloud point which satisfies either of these two criteria will be rejected.

##### Background Point

As shown in Figure 5, the background points might be blocked by the movement of the camera, compared to the object point (which are consistent in different frames). As these background points will introduce the error into the point cloud alignment, they must be removed.

The background points are directly identified from the depth image. Thus, if an abrupt decrease of depth is found to occur in the area nearby a single point, this point is inferred to be a background point. These criteria can be expressed by Equation (Equation 12), where z(u,v) represents the depth value *z* value of the point $p={\left(u,v,z,i\right)}^{T}$ at the (u,v) position and $\pi_{sh,bg}$ is a threshold. Notably, relevant background points tend to exist near the edges of an object; this means that comparing z(u,v) with that of the point which is four pixels away is the most effect means of background point removal. In addition, $\pi_{sh,bg}$ is relevant to the detection range of the selected camera (see Section 5
Table 3).
(12)$$\begin{array}{c} z\left(u,v\right)-z\left(u+4,v\right)>\pi_{sh,bg}*z\left(u,v\right)\;\;\;\;\vee \\ z\left(u,v\right)-z\left(u-4,v\right)>\pi_{sh,bg}*z\left(u,v\right)\;\;\;\;\vee \\ z\left(u,v\right)-z\left(u,v+4\right)>\pi_{sh,bg}*z\left(u,v\right)\;\;\;\;\vee \\ z\left(u,v\right)-z\left(u,v-4\right)>\pi_{sh,bg}*z\left(u,v\right)\;\;\;\;\;\; \end{array}$$

##### Motion Induced Outlier

The initial coordinate transformation $T^{0}$ can be predicted by the IMU integration process (which will be introduced in the next section) or using the transformation of previous frame. Together with the projection model of the camera Equations (Equation 2)–(Equation 4), the predicted positions of the points in the new frame can be calculated from the transformation of the source point cloud $P_{s}$ by Equation (Equation 13). If this predicted point lays out of the FOV (Field-of-View) of the camera (Equation (Equation 14)), it will be removed before the ICP alignment. Note that this step is implemented at the step of making point pairs.
(13)$$p={\left(u,v,z,i\right)}^{T}=\pi^{-1}\left(T^{0}\cdot P_{s}\right)$$
(14)$$\begin{array}{cc} \; & u_{predict}<0\;\vee \;u_{predict}>u_{max}\;\left(i.e.,width\right)\;\;\vee \\ \; & v_{predict}<0\;\vee \;v_{predict}>v_{max}\;\left(i.e.,height\right) \end{array}$$

#### 3.2.2. Acception Rules

Three kinds of acceptance rules are further applied: gradient-based criteria, depth extreme points and canny features. Any point satisfies one of these three rules will be accepted. The relationship of the rejection rule and acceptance rule is ”and,∧”, which means that a salient point needs to satisfy the acceptance rules and it is not a rejected point.

##### Gradient-Based Criteria

The position where the Depth or Intensity changed dramatically are believed to carry the important position information or features. Thus, the criteria based on the intensity gradient (Equation (Equation 15)) and the depth gradient (Equation (Equation 16)) gradient are developed.
(15)$$\begin{array}{cc} \; & \left|i(u+2,v)-i(u-2,v)\right|>\pi_{sh,i}\;\;\;\vee \\ \; & \left|i(u,v+2)-i(u,v-2)\right|>\pi_{sh,i} \end{array}$$
(16)$$\begin{array}{cc} \; & \left|z(u+2,v)-z(u-2,v)\right|>\pi_{sh,z}*z(u,v)\;\;\vee \\ \; & \left|z(u,v+2)-z(u,v-2)\right|>\pi_{sh,z}*z(u,v) \end{array}$$

##### Depth Extreme Point

As shown in Figure 5, the depth extreme points are considered to contain 3D features and thus need to be select as salient points. The extreme point of the depth (the local maximum of minimum of the depth) on a continuous plane can be found by a zero depth gradient $g_{u}\left(u,v\right)=\partial \left(u,v\right)/\partial u=0$ or $g_{v}\left(u,v\right)=\partial \left(u,v\right)/\partial v=0$. As the cloud points are discrete, the extreme points are detected by calculating monotonically the gradient in an interrogation window of 5 × 5 pixels. Thus, the points which satisfy Equations (Equation 17) and (Equation 18) are considered to be the extreme points in the *u* direction. Similar procedures are conducted to extract the extreme points in the *v* direction.
(17)$$\begin{array}{c} g_{u}\left(u,v\right)=z\left(u+1,v\right)-z\left(u,v\right) \end{array}$$
(18)$$\begin{array}{c} (g_{u}\left(u-2,v\right)<0\;\wedge \;g_{u}\left(u-1,v\right)<0\;\wedge \\ \;g_{u}\left(u,v\right)>0\;\;\;\;\;\;\;\wedge \;g_{u}\left(u+1,v\right)>0)\;\;\;\;\;\;\;\vee \;\;\;\\ (g_{u}\left(u-2,v\right)>0\;\wedge \;g_{u}\left(u-1,v\right)>0\;\wedge \\ \;g_{u}\left(u,v\right)<0\;\;\;\;\;\;\;\wedge \;g_{u}\left(u+1,v\right)<0) \end{array}$$

##### Canny Features

The Canny edge detector is believed to be one of the most popular edge detection methods ever since it was developed [25]. It has been shown that the introduction of the canny feature will increase the alignment accuracy. Therefore, this detector is applied to the NIR image to extract the edge points as a supplement of salient point detector. Once the canny edge is detected, the corresponding points in the point cloud will be accepted. The Canny detector form OpenCV with the parameter: threshold1=150, threshold2=300 and apertureSize=3 are used in this work.

### 3.3. Weighting of the Point Pairs

In Kerl’s work [15], intensity residuals are found to follow the t-distribution approximately. We measured the position error distribution of the point pairs from two images. The results of three different cases are shown in Figure 6. In first case, the camera is fixed and the point cloud is captured twice. The second and third cases show the position error distributions of the point clouds by a moving camera with and without running four ICP loops, respectively. As shown in Figure 5, the t-distribution approximates the error distribution better than the normal distribution in every case.

Base on our observation of distribution, we modify the error matrix (Equation (Equation 9)) with the robust weighting factor:(19)$$\Delta T=\underset{\Delta T}{\mathrm{argmin}}\sum_{i}^{size(P)}w\left(i\right)||P_{i}-\Delta T\cdot P_{i}^{\prime }\left|\right|$$
(20)$$w\left(i\right)=\frac{\nu +1}{\nu +{\left(\frac{\left|\right|P_{i}-T\cdot P_{i}^{\prime }\left|\right|-u}{\sigma }\right)}^{2}}$$

In Equation (Equation 19), ν is the degree of freedom. As indicated in Figure 6, ν is close to 2 when a large motion occurs and the alignment of two frames is moderate; while ν increases when the alignment is improved. In the following experiments, ν=4 is set for the weight factor calculation. Assuming the point cloud are well aligned, the mean of the error matrix *u* is set to zero. The deviation can then be calculated by the following equation recursively:(21)$$\sigma =\frac{1}{n}\sum_{i=1}^{n}\left|\right|P_{i}-T\cdot P_{i}^{\prime }\left|\right|\frac{\nu +1}{\nu +{\left(\frac{\left|\right|P_{i}-T\cdot P_{i}^{\prime }\left|\right|}{\sigma }\right)}^{2}}$$

## 4. Data Fusion

### 4.1. Modelling Equations of IMU Sensors

The MEMS IMU sensor consists of a 3-axis gyroscope and a 3-axis accelerometer. The sensor measurements of the angular velocities by the gyroscope $\omega_{m}={\left(\omega_{x},\omega_{y},\omega_{z}\right)}^{T}$ and the accelerations by the accelerometer $a_{m}={\left(a_{x},a_{y},a_{z}\right)}^{T}$ can be described by the following model:(22)$$\begin{array}{cc} \; & \omega_{m}=\omega_{real}+\omega_{b}+\omega_{n}\\ \; & a_{m}=R\left(q\right)\left(a_{real}-g\right)+a_{b}+a_{n}\\ \; & \omega_{n}∼N\left(0,\sigma_{\omega }^{2}\right)\;\;\;\;\;\;\;\;\;\;\;{\dot{\omega }}_{b}=\omega_{bn}∼N\left(0,\sigma_{\omega_{b}}^{2}\right)\\ \; & a_{n}∼N\left(0,\sigma_{a}^{2}\right)\;\;\;\;\;\;\;\;\;\;\;\;{\dot{a}}_{b}=a_{bn}∼N\left(0,\sigma_{a_{b}}^{2}\right) \end{array}$$
where $\omega_{real}$ and $a_{real}$ represent the true angular velocity and acceleration, respectively; $\omega_{n}$ and $a_{n}$ refer to the additive noises of the sensor which follow the Gaussian distributions in nature; and the bias part $\omega_{b}$ and $a_{b}$ can be described as random walk processes. The derivatives of the gyroscope bias $\omega_{bn}={\dot{\omega }}_{b}$ and the accelerometer bias $a_{bn}={\dot{a}}_{b}$ also follow the Gaussian distributions. The error variance $\sigma_{\omega }^{2}$ and $\sigma_{a}^{2}$ can be found in the datasheet of the IMU. Some high precision IMUs also provide the $\sigma_{\omega_{b}}^{2}$ and $\sigma_{a_{b}}^{2}$. If not, this two values can be derived from the IMU calibration process or one can set them as the squares of additive noises. $q={\left(q_{w},q_{x},q_{y},q_{z}\right)}^{T}$ is the quaternion representation of rotation from the inertial frame to the IMU frame. R(q) is the rotation matrix corresponding to the quaternion vector.

### 4.2. Error State Kalman Filter

The Error State Kalman filter (ESKF) is adopted to estimate the errors in every state by using the differences between the IMU data and the Frame to Frame Alignment results. Figure 7 shows the workflow of the proposed estimator. When an IMU data are fed in, the integration process will integrate the nominal state to provide a prediction pose for the frame to frame alignment module. At the same time, the error state will also be updated according to the ESKF processing model. After finishing the alignment process, the output of the alignment module will serve as the measurement to correct the error state. Finally, the new state will be calculated by the composition of the error state and the nominal state.

In the current work, the quaternion (rotation) q, position p, velocity v, together with the biases of the gyroscope $\omega_{b}$ and accelerometer $a_{b}$ are used to describe the system. Following the notation style of Santamaria-Navarro et al. [26], The total system state is described by the true state $x_{t}=\left(q_{t},p_{t},v_{t},{\omega_{b}}_{t},{a_{b}}_{t}\right)$, nominal state $x=(q,p,v,\omega_{b},a_{b})$ and error state $\delta x=(\delta \theta ,\delta p,\delta v,\delta \omega_{b},\delta a_{b})$. In the error state, the Euler angles are selected to represent the rotation so that the dimension of the rotation component is 3. The composition of the state follows:(23)$$\begin{array}{cc} \; & q_{t}=\delta q⊗q,\;\;\;\;\;\;\;\;\delta q=q\left(\delta \theta \right)=e^{\delta \theta /2}\\ \; & p_{t}=p+\delta p,\;\;\;\;\;\;\;\;v_{t}=v+\delta v\\ \; & {\omega_{b}}_{t}=\omega_{b}+\delta \omega_{b},\;\;{a_{b}}_{t}=a_{b}+\delta a_{b} \end{array}$$
where ⊗ indicates quaternion multiplication. In the ESKF fusion, the inputs are divided into 3 parts: u,n and z. The measurement input u contains the readout from the gyroscope and the accelerometer $u=(\omega_{m},a_{m})$. The noise impulse input n is the assumed IMU noise $n=(\omega_{n},a_{n},\omega_{bn},a_{bn})$ and the observation z includes the rotation and translation from the ICP alignment process z=(q,t). According to the IMU measurement module, every component in the noise impulse input follows the Gaussian distribution with a zero mean value. The purpose of doing so is intended to separate the error part from other parts and let the error remain in the error state. The storage in the nominal state is always the best fusion result. The ESKF workflow can be divided into two steps: (1) the ESKF updating process driven by the IMU and (2) the ESKF innovation step from the IMU (see Figure 7), detailed as follows:(1)Update the nominal state through the nominal state processing model $x_{n}=f_{ns}\left(x_{t-1},u_{t}\right)$ and update the error state and the covariance matrix through the error state processing model $\delta x_{n}=f_{es}\left(x_{t-1},u_{t},n_{t}\right)$(2)Innovate the Kalman Gain *K*, the error state and the covariance matrix, and then inject the error state to the nominal state

### 4.3. ESKF Updating Process

#### 4.3.1. Kinematics of True and Nominal State

The system kinematics of the IMU can be describe by the equation:(24)$$\begin{array}{cc} \; & {\dot{q}}_{t}=\frac{1}{2}q_{t}⊗\omega_{t}=\frac{1}{2}q_{t}⊗\left(\omega_{m}-\omega_{bt}-\omega_{n}\right)\\ \; & {\dot{p}}_{t}=v_{t},\;\;\;\;\;\;\;\;{\dot{v}}_{t}={R_{t}}^{-1}\left(a_{m}-a_{bt}-a_{n}\right)+g\\ \; & {\dot{\omega }}_{bt}=\omega_{bn},\;\;\;\;{\dot{a}}_{bt}=a_{bn} \end{array}$$
where q⊗ω represent $q⊗{\left[0,\omega \right]}^{T}$ and the multiplication operation of the quaternion and the angular velocity can be calculated by $q_{t}⊗\omega_{t}=\Omega \left(\omega_{t}\right)q_{t}$ where Ω(ω) is the quaternion integration matrix:(25)$$\Omega \left(\omega \right)=\left[\begin{array}{cccc} 0 & -\omega_{x} & -\omega_{y} & -\omega_{z}\\ \omega_{x} & 0 & \omega_{z} & -\omega_{y}\\ \omega_{y} & -\omega_{z} & 0 & \omega_{x}\\ \omega_{z} & \omega_{y} & -\omega_{x} & 0 \end{array}\right]$$

The nominal state kinematics can be updated following the system module. The noise of the IMU is not considered in this nominal state module, i.e., the nominal state is only updated by the measurement input $u_{t}=\left(\omega_{m},a_{m}\right)$. The kinematics in continuous time is shown as follows:(26)$$\begin{array}{cc} \; & \dot{q}=\frac{1}{2}q⊗\left(\omega_{m}-\omega_{b}\right)=\frac{1}{2}\Omega \left(\omega_{m}-\omega_{b}\right)q\\ \; & \dot{p}=v,\;\;\;\;\;\;\;\;\;\;\dot{v}={R(q)}^{-1}\left(a_{m}-b_{a}\right)+g\\ \; & {\dot{\omega }}_{b}=0,\;\;\;\;\;\;\;\;{\dot{a}}_{b}=0 \end{array}$$

In a discrete time, the nominal state can be calculated by:(27)$$\begin{array}{cc} \; & q\leftarrow q⊗q\{\left(\omega_{m}-\omega_{b}\right)\Delta t\}\\ \; & p\leftarrow p+v\Delta t+\frac{1}{2}\left(R\left(a_{m}-a_{b}\right)+g\right){\Delta t}^{2}\\ \; & v\leftarrow v+(R\left(a_{m}-a_{b}\right)+g)\Delta t\\ \; & \omega_{b}\leftarrow \omega_{b},\;\;\;\;\;\;a_{b}\leftarrow a_{b} \end{array}$$

#### 4.3.2. Updating Process of Error State and Covariance Matrix

The error state kinematics can be described by the following equations.
(28)$$\begin{array}{cc} \; & \delta \dot{\theta }=-R\delta \omega_{b}-R\omega_{n}\;\;\;\;\;\;\;\;\;\;\delta \dot{p}=\delta v\\ \; & \delta \dot{v}=-{\left[R\left(a_{m}-a_{b}\right)\right]}_{\times }\delta \theta -R\delta a_{b}-Ra_{n}\\ \; & \delta {\dot{\omega }}_{b}=\omega_{bn}\;\;\;\;\;\;\;\;\;\;\;\;\;\;\;\;\;\;\;\;\;\;\;\;\;\delta {\dot{a}}_{b}=a_{bn} \end{array}$$

The equations of $\delta \dot{p}$, $\delta {\dot{\omega }}_{b}$ and $\delta {\dot{a}}_{b}$ can be derived directly by the definition of the error state. The derivations of the error velocity $\delta \dot{v}$ and the error orientation $\delta \dot{\theta }$ can be found out in the Appendix B. The error state in the discrete time domain follows:(29)$$\begin{array}{cc} \; & \delta \theta \leftarrow \delta \theta -R\delta \omega_{b}\Delta t-R\omega_{n}\;\;\;\;\;\;\delta p\leftarrow \delta p+\delta v\Delta t\\ \; & \delta v\leftarrow \delta v+\left(-{\left[R\left(a_{m}-b_{a}\right)\right]}_{\times }\delta \theta -R\delta a_{b}\right)\Delta t+Ra_{n}\\ \; & \delta \omega_{b}\leftarrow \delta \omega_{b}+\omega_{bn},\;\;\;\;\;\;\;\;\;\;\;\;\;\;\;\;\;\;\;\;\delta a_{b}\leftarrow \delta a_{b}+a_{bn} \end{array}$$

By applying $n={\left[\omega_{n},a_{n},\omega_{bn},a_{bn}\right]}^{T}$ as the input, which drives the system forward and induces the system transition matrix F and input matrix $F_{i}$, The kinetics of the error state and covariance of the error state can be represented by:(30)$$\begin{array}{cc} \; & \delta x_{t}=F\delta x_{t-1}\\ \; & \Sigma =F\Sigma F^{T}+F_{i}Q_{imu}F_{i}^{T}\\ \; & F=\left[\begin{array}{ccccc} I & 0 & 0 & F_{1} & 0\\ 0 & I & F_{2} & 0 & 0\\ F_{3} & 0 & I & 0 & F_{1}\\ 0 & 0 & 0 & I & 0\\ 0 & 0 & 0 & 0 & I \end{array}\right]F_{i}=\left[\begin{array}{cccc} I & 0 & 0 & 0\\ 0 & 0 & 0 & 0\\ 0 & I & 0 & 0\\ 0 & 0 & I & 0\\ 0 & 0 & 0 & I \end{array}\right]\\ \; & F_{1}=\left[\begin{array}{c} -R\Delta t \end{array}\right]\;\;\;\;\;\;\;\;\;\;F_{2}=\left[\begin{array}{c} I\Delta t \end{array}\right]\\ \; & F_{3}=\left[\begin{array}{c} -{\left[R\left(a_{m}-a_{b}\right)\right]}_{\times }\Delta t \end{array}\right] \end{array}$$

Here ${\left[•\right]}_{\times }$ is the skew operator which produces the skew-symmetric matrix. According to the IMU measurement model, the covariance matrix $Q_{imu}$ of the the perturbation input $n={\left[\omega_{n},a_{n},\omega_{bn},a_{bn}\right]}^{T}$ is: (31)$$Q_{imu}=diag\left(\sigma_{\omega }^{2}{\Delta t}^{2},\sigma_{a}^{2}{\Delta t}^{2},\sigma_{\omega_{b}}^{2}\Delta t,\sigma_{a_{b}}^{2}\Delta t\right)$$

### 4.4. ESKF Innovation Process

In Figure 6, the output of ICP alignment is the orientation q and the position p of the camera in the world frame. It is necessary to transfer the pose to the IMU link through Equation (Equation 32):(32)$$T_{WI}=T_{WC}\cdot T_{IC}^{-1}$$
where $T_{WC}$ is the ICP result representing the transformation of Camera in the world frame, $T_{IC}=\left[R_{IC}|t_{IC}\right]$ is the installation geometry of camera in the IMU link (see Figure 1) and $T_{WI}=\left[R_{WI}|t_{WI}\right]$ is the IMU pose in the world frame. $T_{IC}$ in the current setup is represented by Equation (Equation 1). $T_{WI}$ is then used as the input of the filter $y={\left[\theta ,p\right]}^{T}$. Since the estimate of the error state is zero ($\bar{\delta x}=0$), the innovation z and the covariance matrix Σ are:(33)z=y−h(x)
(34)$$\Sigma =H\Sigma H^{T}+Q_{icp}$$
where h(x) is the measurement model of the system state, H is the jacobian matrix of the measurement model and $Q_{icp}$ is the covariance of ICP. As the orientation and the position are directly measured by ICP, H is the identity matrix accordingly. Therefore, the Kalman gain, the estimate of the error state and the covariance matrix can be innovated.
(35)$$\begin{array}{cc} \; & K\leftarrow \Sigma H^{T}\Sigma^{-1} \end{array}$$
(36)$$\begin{array}{cc} \; & \bar{\delta x}\leftarrow Kz \end{array}$$
(37)$$\begin{array}{cc} \; & \Sigma \leftarrow \Sigma -KZK^{T} \end{array}$$

After the innovation process is done, the error state is then injected into the nominal state following the general composition rules defined in Equation (Equation 23). The injection process make sure that the nominal state is always updated. Afterward the error state will be reset to zero (δx=0) and the orientation part of the covariance matrix need to be updated according to the newest nominal state, as follow: (38)$$\Sigma_{\delta \theta }\leftarrow G\Sigma_{\delta \theta }G^{T}$$
where G is the Jacobian matrix of the nominal state toward the error state and
(39)$$G=I+{\left[\frac{1}{2}\delta \theta \right]}_{\times }$$

### 4.5. Integration and Re-Integration

In the VIO system, the IMU data are updated at a high rate (i.e., >200 Hz for a typical MEMS IMU) while the camera capture rate is relatively slow (i.e., <30 Hz). In addition, the ICP processing time is dependent on the numerical convergence rate and is not constant. Therefore, we need to maintain the state queue and update it properly using the integration/re-integration method.

As shown in Figure 8, the above procedure results in a fixed-length state queue being maintained. The integration of the state is driven by the IMU measurement, i.e., whenever an IMU measurement is inputted, the state will be updated and stored in the queue. When the ICP and filter innovation processes are concluded, the state will be updated according to the capture timestamp. Then, the re-integration process integrates the state from the innovation state to the most recent state. As a result, the system state is always updated and the VIO output rate can keep pace with the IMU update rate.

## 5. Results and Discussion

Some threshold parameters are relevant to the camera resolution and its detection range (Table 3). Also, the sampling rates of different sensors and camera calibration parameters are listed in Table 4.

Due to there is no existing public ToF-IMU dataset, we first conducted the handheld test and UAV test and compared the results with those of the motion capture system, which serve as the ground truth benchmarks. The accuracy of the odometry is presented by the root mean square error (RMSE) of translational drift. Both the absolute trajectory error (ATE) and the one-step relative pose error (RPE) cases are considered [27]. The definitions of ATE and RPE are shown below: (40)$$\begin{array}{cc} \; & E_{ATE,i}=T_{gt,i}^{-1}ST_{est,i} \end{array}$$(41)$$\begin{array}{cc} \; & E_{RPE,i}={\left(T_{gt,i}^{-1}T_{gt,i+1}\right)}^{-1}\left(T_{est,i}^{-1}T_{est,i+1}\right) \end{array}$$(42)$$\begin{array}{cc} \; & RMSE\left(E_{1:n}\right):={\left(\frac{1}{n}\sum_{i=1}^{n}{∥trans(E_{i})∥}^{2}\right)}^{\frac{1}{2}} \end{array}$$
where $T_{gt,i}$ is the transformation of ground truth of the frame *i*, $T_{est,i}$ the estimate transformation of the frame *i* and S the least-squares solution that maps the estimated trajectory $T_{est,1:n}$ onto the ground truth trajectory $T_{gt,1:n}$. We then carried out a UAV field test and an exploration test on a ground moving platform. The results were compared with those obtained by RealSense T265 VIO sensors. All data is provided in rosbag and are compatible with the TUM dataset [28].

### 5.1. Handheld Test

In the handheld test, we held the VIO platform and successively moved in the x, y, and z directions. This generated a good agreement between estimated trajectory and ground truth, as can be observed in Figure 9 and Figure 10. The length of the trajectory is 12.86 m and ATE and RPE of the estimated trajectory is 0.047 m and 0.017 m/s, respectively.

Based on this dataset, Figure 11 shows the comparison of ground truth trajectory with those using algorithms of the conventional ICP (**icp**), the ICP with the salient point selection criteria (**icp+s**), the ICP with the salient point selection criteria and robust weighting factor (**icp+s+w**), the random sub-sampled based ICP (**icp+rd**) and the ICP with the RANSAC pipeline (**icp+r**) [29]. The corresponding accuracy and processing time of each algorithm are also evaluated and listed in Table 5. It can be seen that the conventional ICP workflow can provide the accurate pose estimation but the processing time is unfavorable to the real-time applications. Both the random sub-sampling based ICP and the ICP with our salient point selection method can dramatically reduce processing time. Nevertheless, the former induced a large ATE error. The ICP with the RANSAC method achieves best in the RPE error but with the largest ATE error and a second largest average processing time. Notably, the conventional ICP, **icp+s**, **icp+r**, and **icp+rd** are all with weights for points set to equal. Together with the robust weighting factor (the weights for points set to the t distribution), the proposed method (**icp+s+w**) in this work reaches almost the same ATE accuracy and 26% improvement in RPE accuracy when compared with the conventional ICP. In the meanwhile, the processing time of icp+s+w is only 25% that of the conventional ICP. The superiority of icp+s+w over other methods is clearly illustrated.

Furthermore, we visualize the salient points in Figure 12a. Our selection algorithm selects the feature and the edge of the object from the raw input point cloud. The number of salient points is consistent (Figure 12b) even if the size of input cloud varies. Together with the good accuracy of icp+s and icp+s+w methods, it is believed that the salient point selection criteria is efficient and robust.

Figure 13 demonstrates that the error of mean $\|p-p^{{}^{\prime }}\|$ in each frame (different lines) decreases with the increase of ICP loops. In general, three (no motion) to 15 ICP loops were performed and the motion prediction of IMU provided the initial guess of the ICP algorithm. After about ten loops, the mean $\|p-p^{{}^{\prime }}\|$ almost reach the minimum value. The refinement process continues until the corresponding transformation change (ΔT) is relatively small or the maximum loops (15 in our experiment) are reached.

### 5.2. UAV Test in LAB Environment

In the UAV test, we mounted the VIO system on our UAV platform and flew the UAV in a circular trajectory (13.019 m) in a laboratory, generating the data in Figure 14 and Figure 15, with an ATE 0.04 m and RPE of 0.021 m/s. Note that we turned off the ambient light during the test, which induced the failure of the mono camera; however, the ToF camera based VIO system continued to function in this dark environment (Figure 16). The full video of this test can be found in Appendix A.

### 5.3. Field Test in Corridor

We mounted the ToF–VIO sensor and a Realsense T265 sensor on the UAV platform and conducted a field test in a corridor. The UAV took off, flew straight along the corridor, and landed 5 m in front of the takeoff position. The trajectory length including takeoff and landing is 8.23 m. Figure 17 presents the comparison between the estimated trajectory generated by the current ToF–VIO sensor and that by the Realsense T265 sensor. In general, the agreement is good, especially in the early period. The ATE and RPE between ToF–VIO and T265 is 0.12 m and 0.029 m/s, respectively. As the UAV took off and landed at the same ground level, our ToF–VIO has better estimation in the *z*-direction.

### 5.4. Exploration Test Using a Ground Moving Platform

To further demonstrate the performance of our system in a longer range, we mounted the ToF–VIO sensor and a Realsense T265 sensor on a ground moving platform to explore in the indoor Lab environment. The Realsense T265 sensor were used as the benchmark. The trajectory length for this experiment sequence is 25.675 m (captured by T265 sensor). Figure 18 shows the reconstructed images of the lab and the corresponding trajectories by ToF–VIO and T265. The colors of these points represent the z values (height). As seen, the map shows many detailed features, including a flat ground and straight walls. Figure 19 presents the comparison between the estimated trajectory generated by these two sensors. Good agreement between these two sensors can also be observed. The ATE and RPE between our ToF–VIO and T265 is 0.78 m and 0.025 m/s respectively. Compared with the UAV field test in Section 5.3, the ATE over the trajectory length increase from 1.4% to 3.0%. As known, in an unknown environment without any reference map, the ATE will be accumulated as the drift is not compensated. However, the PRE of these two tests remain at the same level (<0.03 m/s), even when the range increases. The pose estimations from these two sensors agree with each other. The full video of this test can be found in Appendix A.

### 5.5. Analysis

As shown in Figure 20, the number of salient points is only 5% of the number in the original point clouds. Aside from small variations, the number of salient points remains consistent in different frames. We can therefore efficiently obtain a pose estimation by alignment of these points.

We tested our algorithm on a TX2-embedded computer and an Intel i5 PC, and the resulting calculation times are listed in Table 6. The results show that our algorithm can estimate the pose at a rate of at least 15 Hz, which means it can be used in realtime applications.

## 6. Conclusions

In this paper, we have described the development of a ToF camera-based VIO system. This system was demonstrably superior to the conventional ICP-based workflow, as the computational time was reduced by salient-point selection criteria and the robustness of frame-to-frame alignment was ensured by the statistic weight function. The IMU data were loosely coupled in the proposed system with an ESKF to provide the ego-motion pose estimation. We then assembled our experimental platform and conducted a field test. The results showed that our proposed approach achieved similar accuracy to the state-of-the-art VIO system. Experimental data also showed that our system exhibited excellent performance in an environment of varying ambient-light intensity and in a totally dark environment. The limitation of this study is the range of the ToF camera. The depth detection range of the current ToF camera is 4 m, which is short and will limit the vehicle speed and operation time for the mission. With the development of ToF sensor technology, this limitation will be relieved and the current algorithm can still find good applications.

## Figures and Tables

**Figure 1 sensors-20-01263-f001:**
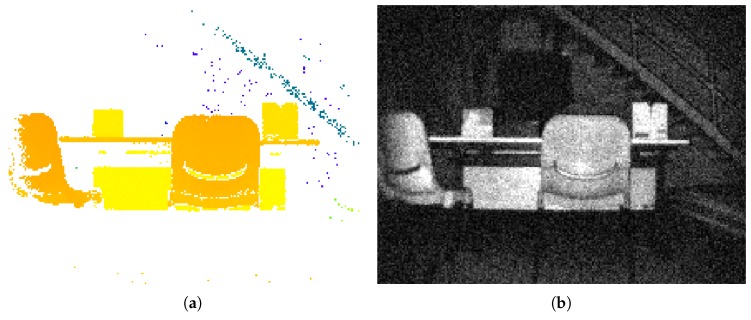
PMD Flexx ToF camera output (224 x 171 pixels). (**a**) Depth image, white color mean depth information is not available. (**b**) NIR intensity image.

**Figure 2 sensors-20-01263-f002:**
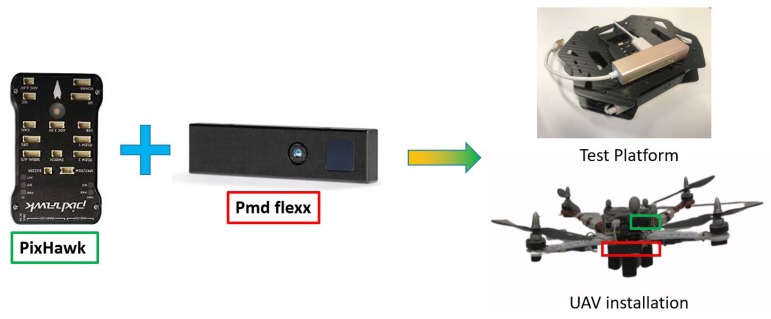
ToF–VIO testing platform and its implementation on a UAV.

**Figure 3 sensors-20-01263-f003:**
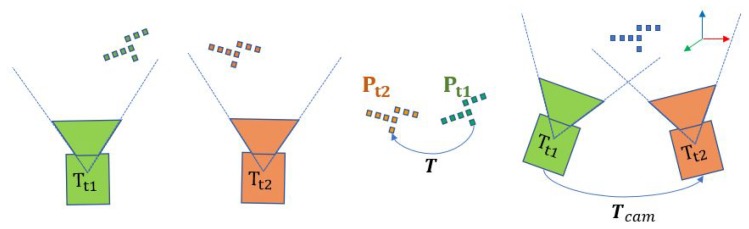
The camera pose in the world frame and the point cloud alignment procedure in the camera frame.

**Figure 4 sensors-20-01263-f004:**
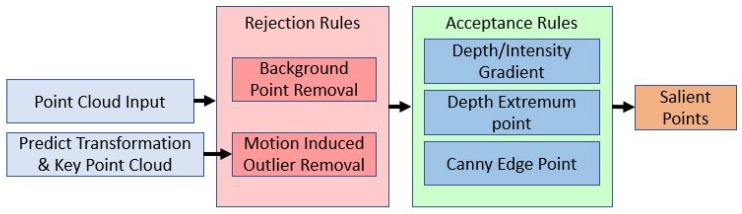
Workflow of sailent point selection.

**Figure 5 sensors-20-01263-f005:**
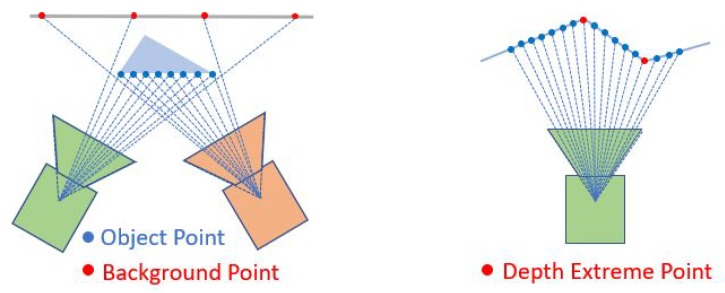
Background points and Depth extreme points.

**Figure 6 sensors-20-01263-f006:**
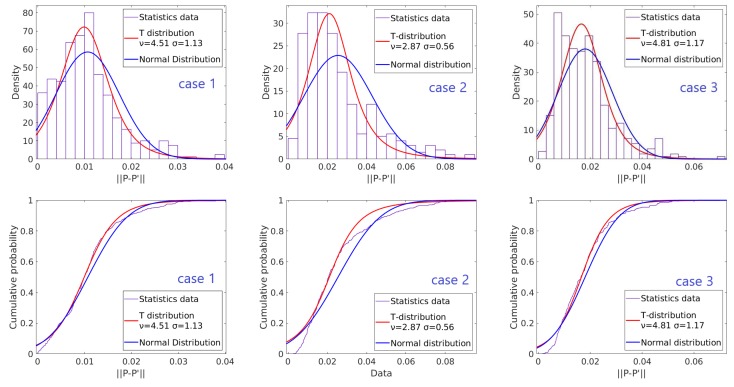
Distribution and approximate distribution curve of the $\left|\right|P_{i}-{P_{i}}^{\prime }\left|\right|$ in different cases. Uper figures plot the probability density function(PDF) and lower figures plot the cumulative distribution function (CDF).

**Figure 7 sensors-20-01263-f007:**
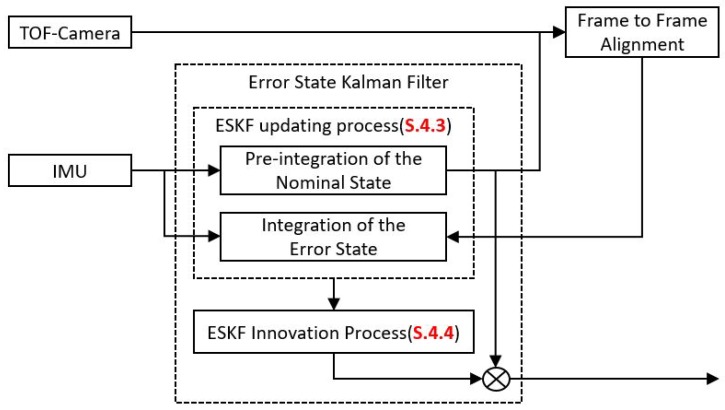
Workflow of the Error State Kalman Filter estimator by using the ToF camera alignment results and the IMU data.

**Figure 8 sensors-20-01263-f008:**
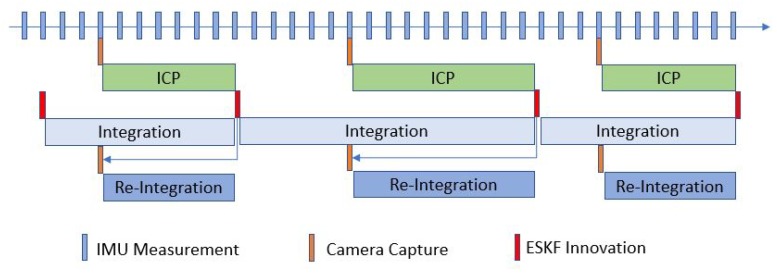
Time synchronization of IMU and ICP.

**Figure 9 sensors-20-01263-f009:**
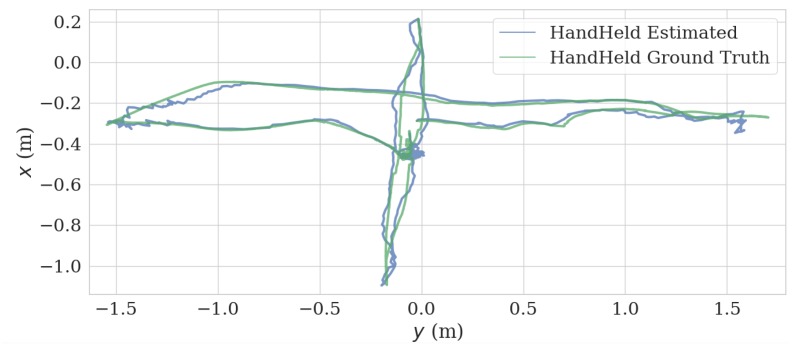
Estimate trajectory and ground truth from *x*–*y* plane of hand held test.

**Figure 10 sensors-20-01263-f010:**
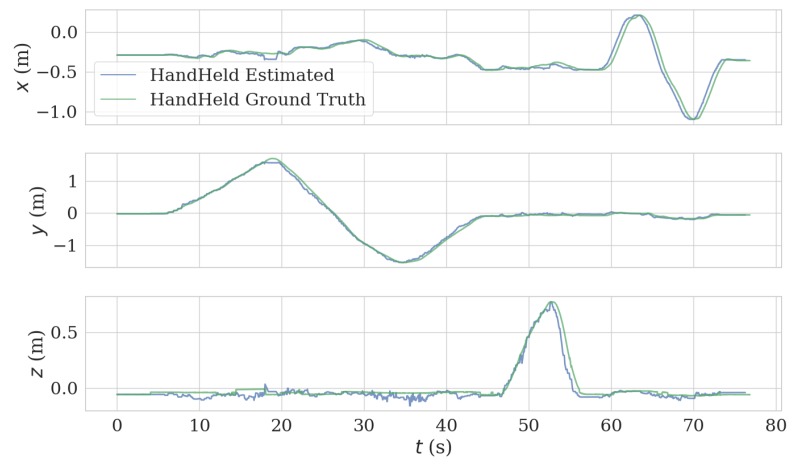
Comparison of estimated trajectory and ground truth of hand held test.

**Figure 11 sensors-20-01263-f011:**
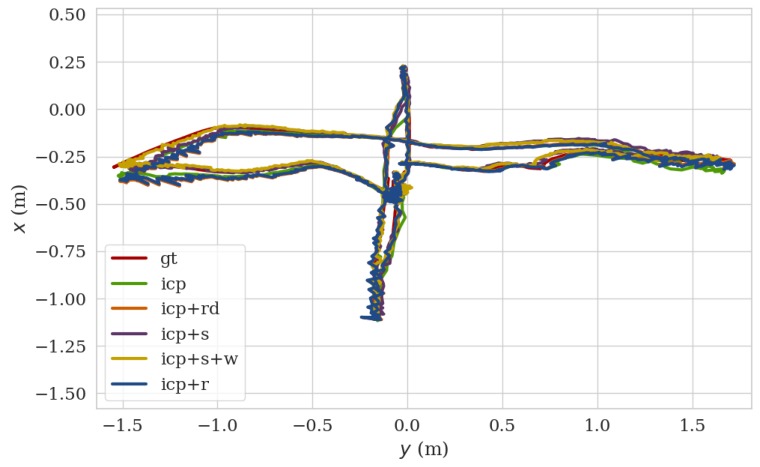
Trajectories of different ICP methods.

**Figure 12 sensors-20-01263-f012:**
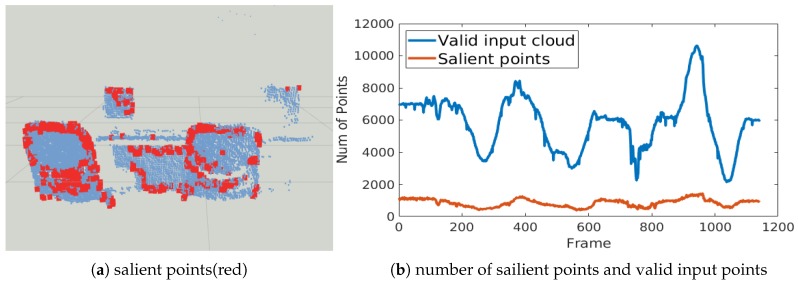
Visualization of salient points.

**Figure 13 sensors-20-01263-f013:**
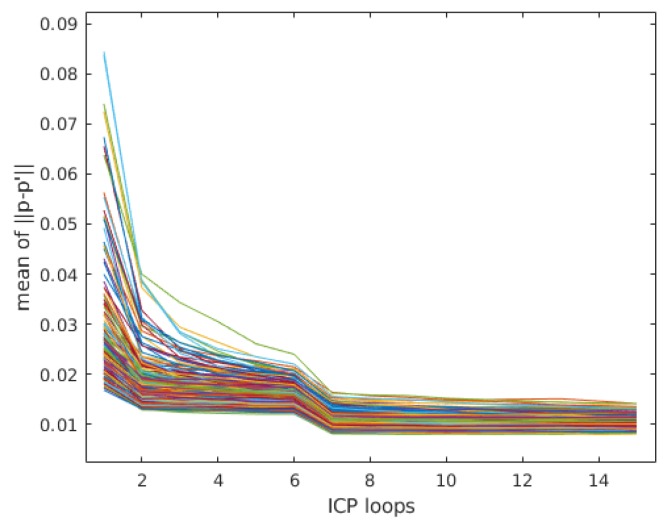
The mean value of $||p-p^{\prime }\left|\right|$ in ICP loops.

**Figure 14 sensors-20-01263-f014:**
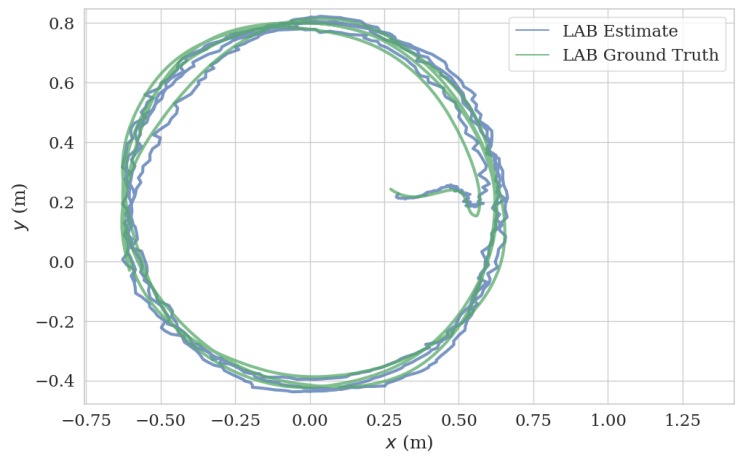
Estimated trajectory and ground truth from an *x*–*y* plane of lab environment test.

**Figure 15 sensors-20-01263-f015:**
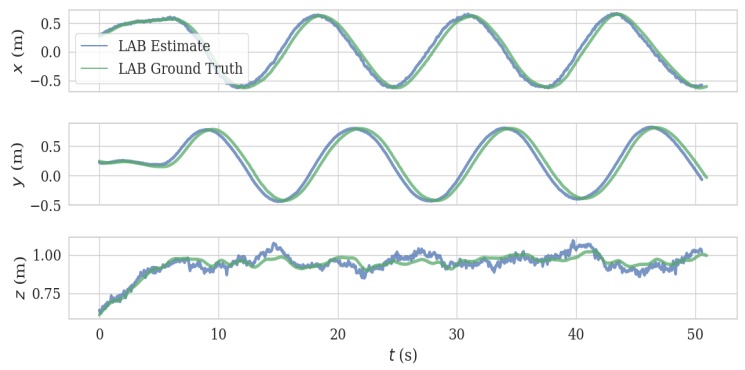
Comparison of estimated trajectory and ground truth of in lab environment test.

**Figure 16 sensors-20-01263-f016:**
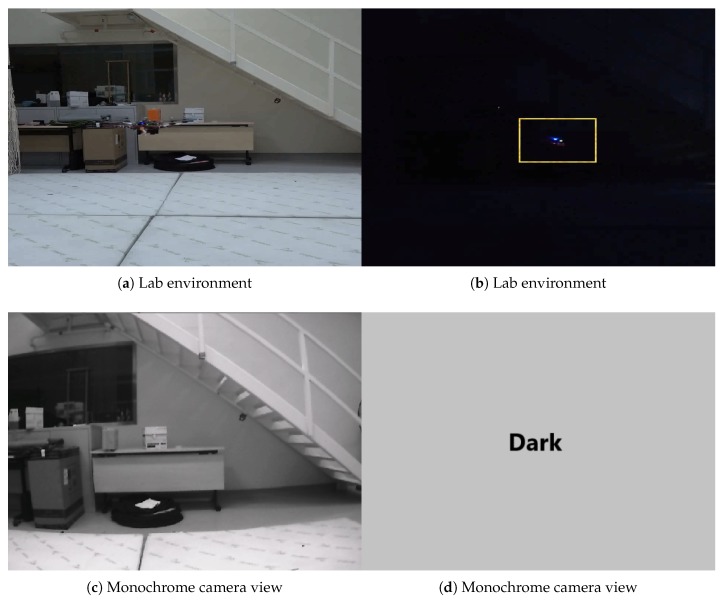
The VIO results obtained in an environment of varying ambient light intensity. (**a**,**c**,**e**,**g**,**i**) are recorded when light is on, (**b**,**d**,**f**,**h**,**i**) are recorded when light is turned off.

**Figure 17 sensors-20-01263-f017:**
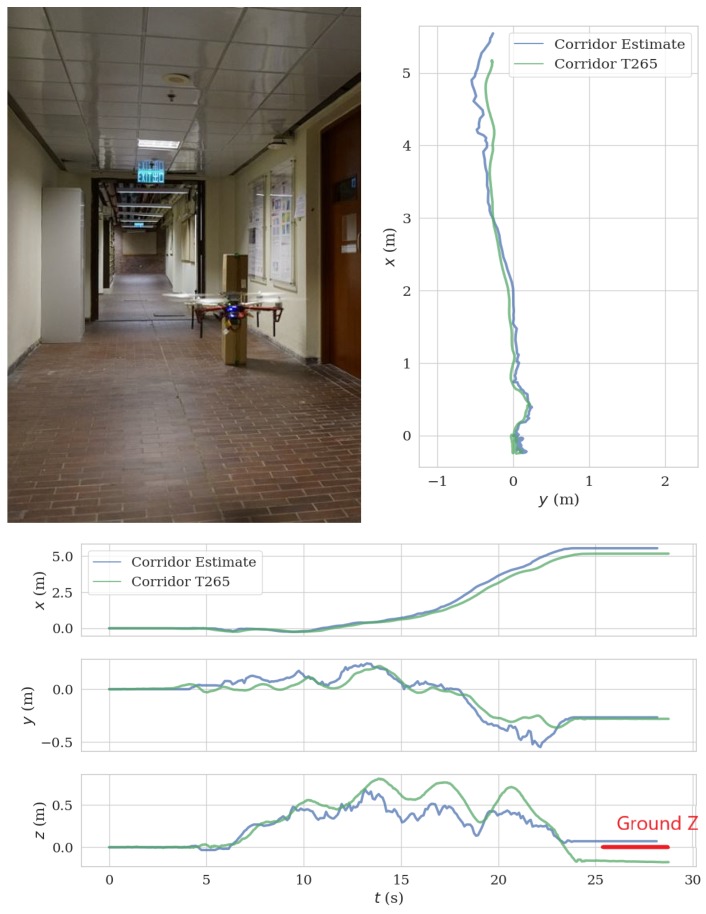
Comparison of the estimated trajectory by the current ToF–VIO sensor with that by the Realsense T265 sensor in the corridor environment.

**Figure 18 sensors-20-01263-f018:**
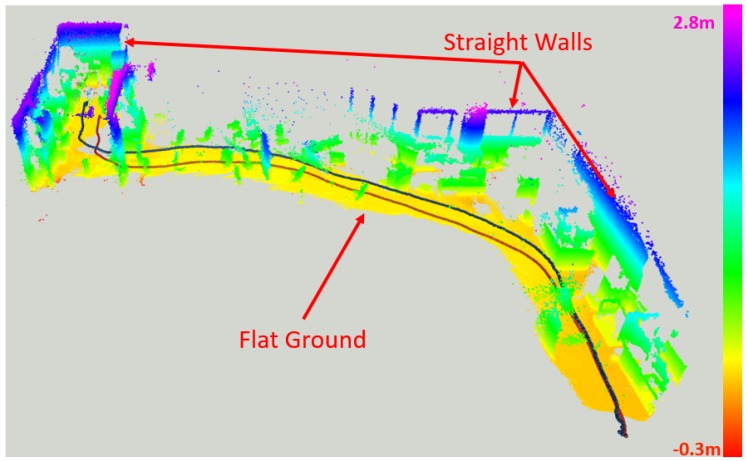
Exploration and restruction of the indoor environment using ToF–VIO, blue path (ToF–VIO), red path (T265).

**Figure 19 sensors-20-01263-f019:**
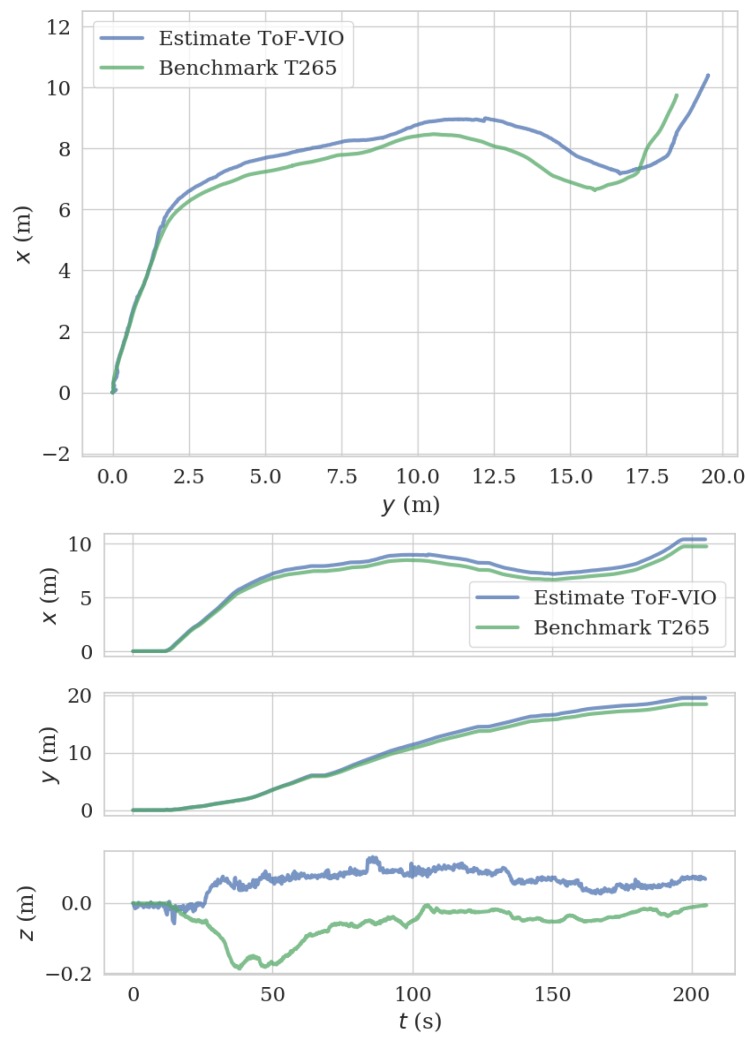
Comparison of the estimated trajectory by the current ToF–VIO sensor with that by the Realsense T265 sensor in the exploration test.

**Figure 20 sensors-20-01263-f020:**
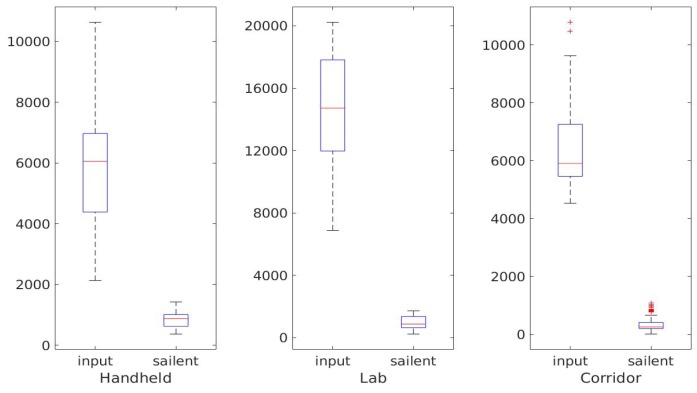
Box plots of the numbers of input points and salient points in three tests.

**Table 1 sensors-20-01263-t001:** Different VIO frameworks.

		IMU Fusion Method	Loose-Coupled	Tight-Coupled
Estimation Method and Input				
Direct method		RGB/Grey	SVO+MSF [3,4]	VINS [7] [8] VI-DSO [10]
Image-based method	Feature based	RGB/Grey	MSCKF [5]	VI-ORB [12] OKVIS [6]
	ICP	Depth + RGB/Grey	KinectFusion [11] (no IMU support)	
		Depth + Grey	(current work)	n.a.
Others	NDT	RGB/Grey + Lidar	3D-NDT [13], Direct Depth SLAM [14]	

**Table 2 sensors-20-01263-t002:** Pros and cons of using passive camera and active ToF camera in motion estimation.

	Passive Camera	Active ToF Camera
pros	Less expensive in price Studied for years and with many CV algorithm Relatively high resolution	Strong resilience to ambient light Sense depth infomation without post-processing
cons	Fail in poor/changing lighting conditions Need initialization process to recover the depth	Limit sensing distance (less than 10 m) Current CV algorithm can not used on the NIR image Relatively low resolution

**Table 3 sensors-20-01263-t003:** Device Related Parameter Selection.

Symbol	Value	Description
$\pi_{sh,bg}$	0.01	Background outlier threshold
$\pi_{sh,i}$	100	Intensity gradient threshold
$\pi_{sh,z}$	0.07	Depth gradient threshold

**Table 4 sensors-20-01263-t004:** Detail of Sensors Information.

Topic Name	Content	Frequency
/image_depth	Depth image	15
/image_nir	8-Bit NIR intensisty image	15
/points	Organized point cloud	15
/camera_info	Camera Information $\left(f_{x},f_{y},c_{x},c_{y}\right)$	15
/imu	IMU data	250
/gt	Ground truth captured by Vicion	50

**Table 5 sensors-20-01263-t005:** Accuracy and processing time comparison of different methods.

Method	Translation error (RMSE)				Average Processing Time (ms)	Relevant Processing Time $\frac{t(x)-t(\mathrm{icp})}{t(\mathrm{icp})}$
	ATE (m)	Improvement $\frac{\mathrm{ATE}(x)-\mathrm{ATE}(\mathrm{icp})}{\mathrm{ATE}(\mathrm{icp})}$	RPE (m/s)	Improvement $\frac{\mathrm{ATE}(x)-\mathrm{ATE}(\mathrm{icp})}{\mathrm{ATE}(\mathrm{icp})}$		
**icp**	0.048	0	0.023	0	148	100%
**icp+s**	0.051	–6%	0.019	17%	28.0	18%
**icp+s+w**	0.047	2%	0.017	26%	36.2	25%
**icp+rd**	0.063	–31%	0.021	9%	20.4	14%
**icp+r**	0.070	–45%	0.015	28%	83.8	56%

**Table 6 sensors-20-01263-t006:** Calculation times of different processes of our algorithms on a Intel-i5 PC and TX2 embedded computers.

Process	Time (TX2)	Time (i5-PC)
Salient Point Selection	6 ms	4 ms
ICP alignment	33 ms	30 ms
ESKF	1 ms	1 ms

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
