# Peer review of "Perception in the Dark; Development of a ToF Visual Inertial Odometry System"

_sensors, 2020, doi:10.3390/s20051263_

Round 1

Reviewer 1 Report

Overall, the article presents a system with frame-to-frame ICP with IMU in the Kalman fusion scheme for UAV applications. I believe almost everything is presented nicely but the system is rather simple when compared to other VO/SLAM solutions and evaluated on short sequences (more is needed in that part of the article). Moreover, I have some concerns:

1. In the introduction, you distinguish between the optimization method and the ICP method. I believe that the wording is unfortunate as ICP is also based on optimization. I suggest direct or image-based optimization for that type of optimization.

2. How many ICP iterations do you perform? I assume that it does have to be much considering the motion prediction with IMU.

3. Your experimental section contains rather short sequences. Can you present the influence of some elements of your system on the finally obtained results? For example, what would be the result if the weights for points for the ICP were set to equal instead of t distribution?

4. What are the lengths and final position errors for the presented sequences? With these values, you could compute error as a percentage of the distance and thus indirectly compare to other systems.

5. I believe "pre-integration" means something different to the community than what is shown in the article. I would say that you just perform "integration" and refer to the "pre-integration" when the linearization points are fixed and the new estimates are corrected based on the linearized terms as proposed by Forster in "IMU Preintegration on Manifold for Efficient Visual-Inertial Maximum-a-Posteriori Estimation". If you agree please change it especially in Fig. 7.

6. The usual ICP solutions consist of frame-to-frame and frame-to-map pipelines with frame-to-map to increase accuracy and reduce drift over time. Did you consider experiments with longer sequences? What would be the impact of lacking frame-to-map ICP when you operate for longer periods in the same environment?

7. Minor issues:
- abbreviations Equ is rather strange and I would recommend Eq.
- there is an additional line in Table 3 that is not needed
- there is once "Jacobin" instead of "Jacobian"

Reviewer 2 Report

The authors present a visual odometry system based on a ToF camera, and IMU and a modified ICP with an EKF system. The ICP is heavily sub-sampling the points based on some criteria and is also using weights for the points. What follows is an ESKF implementation with pre and re integration of the IMU.

The paper addresses an important challenge: visual odometry in dark environments. The proposed method is technically sound, but not terribly innovative. The description of the algorithm and of the experiments is quite good.

The beginning of the paper is quite sloppy - another round of proof reading would have been better. See the comments below. The English could use some improvement.

The experimental evaluation of the algorithm is thorough for this algorithm, but comparisons to other algorithms are missing. The paper needs to be improved in this respect.

Comments:

In your main contributions you do not list the work on ICP with the elaborate sub-sampling and weighing. Why? Did you forget? Or is this previous work (if so you should cite this more clearly!).

How did you find the "installation geometry" of the two sensors? Did you perform an extrinsic calibration?

Eq. 9 and Eq. 19: Traditional ICP is using closed form solutions to calculate the transform given the matched point pairs (e.g. SVD/ Horn's Algorithm) - why are you using such a formulation (Eq. 9) and a recursive algorithm? Explain in the text.

The first sentence is missing its beginning...

in the abstract: "proven accurate": Experiments can never 'prove' a systems performance - there may still be situations in which your system does not work. Re-formulate (e.g. 'showed').

in abstract: "present a realtime visual initial system" should be "inertial"

Fig 12: TOF Camera view: Is this the depth image or the intensity image? In any case, it seems to be binary? (either gray or black). Please improve to show both TOF depth and TOF intensity.

The reviewers experience with ToF cameras is, that they suffer heavily from motion blurr. Please comment.
Also please comment on the "wrap-around" error of ToF cameras: Such cameras typically cannot distinguish when the modulated frequency wraps around - so 1m and 5m distance could not be distinguished? How do you deal with that?

The range of 4m is quite limited for this kind of application. E.g. in Fig. 13 the camera will hardly see anything, because things are too far away!? Please comment on that in the text.

"This system was demonstrably superior to the conventional ICP based workflow," You did NOT compare to a conventional ICP method. We don't know if conventional ICP maybe has a much better accuracy.

You should compare your method to other algorithms, like OKVIS or VINS-Mono on the intensity image, or RGBD versions of it (e.g. https://www.mdpi.com/1424-8220/19/10/2251/htm ) - using your dataset.

For sure, you should compare to a "normal" ICP with sub-sampled point clouds - it should run with comparable speed (but maybe lower accuracy).

Minor comments:

line 33: "the object function." => "the objective function."

"Our sensor platform consists of a pmd technologies Flexx ToF camera and an IMU sensor
embedded in a Pixhawk v2 flight controller, as shown in Figure 1, our sensor platform consists
of a PMDtechnologies Flexx ToF camera and an IMU sensor embedded in the Pixhawk v2 flight
controller." - copy & paste error

Round 2

Reviewer 1 Report

Thank you for the prepared cover letter. 

The only remaining issue that I have is then the length of sequences that is rather short and therefore it is hard to determine the accuracy of your solution. I also believe that the achieved accuracy of the system in the experiment with the ground-truth is great (<1%) because you always keep some last keyframes and thus it works similarly to moving in the known map. As a result, it is not a good measure of the performance of your system when exploring in the new environment.

I believe that in order to publish, you either need to:

  • present results on any other public dataset,
  • compare with another publicly available solution on your sequences
  • present more experiments with T265. 

I believe that the 3rd option requires the least amount of work as you would just need to find a proper environment for experiment with T265 (a lot of visual saliency) and just move for more than 25 meters (i.e. any indoor desk environment would work). Then report the accuracy of the proposed system.

Reviewer 2 Report

Dear authors,

thank you for the detailed reply. And thank you for doing the additional experiments - those make the paper really good - congratulations.

Minor comment:

The section "7. Patents" is empty - remove!?

Round 3

Reviewer 1 Report

It is ok. 3% is not great but it least it shows the overall accuracy that can be expected with your approach. Take one more look for English typos like "bule path" in caption for fig. 18.